# Hepatocellular Carcinoma: Past and Present Challenges and Progress in Molecular Classification and Precision Oncology

**DOI:** 10.3390/ijms241713274

**Published:** 2023-08-26

**Authors:** Philip Coffin, Aiwu He

**Affiliations:** MedStar Georgetown University Hospital, Lombardi Cancer Center, 3800 Reservoir Rd NW, Washington, DC 20007, USA; aiwu.r.he@medstar.net

**Keywords:** hepatocellular carcinoma, precision oncology, systemic therapy, genome

## Abstract

Hepatocellular carcinoma (HCC) is one of the most common solid tumor malignancies in the world and represents roughly 90% of all primary malignancies of the liver. The most common risk factors for HCC include hepatitis B virus, hepatitis C virus, alcohol, and increasingly, fatty liver. Most HCC is diagnosed at advanced stages, excluding the possibility of curative resection, which leaves systemic therapy as the only treatment option. However, given the extreme mutational diversity and heterogenous nature of HCC, efforts to develop new targeted systemic therapies were largely unsuccessful until recently. HCC pathogenesis is thought to be a multistage process driven by a wide array of nonmutually exclusive driver mutations accompanied by many passenger mutations, with the average tumor possessing approximately 40 genomic aberrations. Over the past two decades, several efforts to categorize HCC prognostically and therapeutically according to different molecular subclassifications with the intent to guide treatment and identify drug targets have emerged, though, no single consensus has been reached. Recent breakthroughs in drug development have greatly expanded treatment options, but the ideal of uniting each patient’s unique HCC with a targeted systemic therapy remains elusive.

## 1. Introduction

Hepatocellular carcinoma (HCC) is one of the most common solid tumor malignancies in the world and accounts for a significant share of cancer-related morbidity and mortality, and in the United States, this share is growing [1,2].

Approximately 90% of all primary malignancies of the liver are HCC [3]. The two most common risk factors for HCC globally are chronic viral infection with hepatitis B virus (HBV) and hepatitis C virus, estimated to cause 75–85% and 10–20% of cases, respectively. Together, both viruses account for at least 80% of HCC cases [4,5,6]. Alcohol is also a major risk factor and is the second most common risk factor for HCC in both the United States and Europe [7].

Additionally, nonalcoholic fatty liver disease (NAFLD) and nonalcoholic steatohepatitis (NASH) are becoming more recognized risk factors for HCC development and are increasingly common, particularly in the West [3,8]. The coincidence of common comorbidities in this population, such as obesity (OR = 2.69; CI 95% 1.74–4.25) and diabetes mellitus (OR = 4.19; CI 95% 2.71–6.50), predisposes to greater likelihood of progression to cirrhosis and subsequently the development of HCC [9].

In those diagnosed without metastasis, multifocal tumors, or vascular invasion, curative intent with surgical resection is the treatment of choice [10]. Surgical resection is limited to less than 30% of patients, however, because of frequent multifocal disease and surgical limitations resulting from concurrent cirrhosis [11]. Additional options include locoregional therapies with transarterial chemoembolization (TACE), radiofrequency ablation (RFA), or yttrium-90 (Y90) therapy [12,13]. However, most HCC is diagnosed at advanced stages that exclude the use of these therapies, leaving systemic therapy as the only option [14]. Until 2017, there was only one United States Food and Drug Administration (USFDA)-approved systemic therapy for advanced HCC, sorafenib [15], and prior to 2007, the mainstay of advanced HCC treatment was supportive care. Since 2017, several new first-line and second-line systemic therapies for HCC have come on the market. The reason for the sluggish development of targeted systemic therapies for HCC are largely owed to the fact that HCC is an extremely diverse and heterogenous disease, with dozens of mutations identified and large variability among tumors depending on the etiology of each tumor. Despite so many identified mutated genes, it is estimated that perhaps only 25% of HCC cases have an actional mutation, and most mutations have an overall prevalence of <10% [3].

The purpose of this review is to provide an integrated narrative account of the simultaneous efforts to establish a uniform and validated molecular subclassification system for HCC, with prognostic and therapeutic utility, with the ongoing but separate pursuit of identifying systemic therapeutic agents. Specifically, these molecular subclassifications are attempts to further characterize HCC and describe distinct disease entities based on patterns of mutational, genomic, histological, and clinical differences. Following a brief introduction to the origin and basic principles of precision oncology, this review then discusses the main pathogenic mechanisms of HCC development. Next, a detailed summary of the past two decades of attempts at creating comprehensive molecular subclassification systems for HCC is discussed. Subsequently, this review discusses the impact of these classification systems on guiding therapeutic development of targeted systemic therapies and use of clinical biomarkers and the unique challenges faced regarding HCC that have hindered progress in this expanding field.

## 2. Origins and Basic Developments in Precision Oncology

At the turn of the millennium, the world witnessed a major paradigm shift in the ways by which cancer would be diagnosed and subsequently treated. In 2001, the Human Genome Project published the first fully sequenced human genome, followed shortly thereafter by the publication of the world’s first cancer genome [16]. In the same year, the USFDA granted fast-track approval to imatinib, developed by Novartis and marketed in the United States as Gleevec, for use in patients with chronic myeloid leukemia (CML). This approval stemmed from the data of three large multicentered phase II clinical trials demonstrating miraculous results by which the vast majority of patients with chronic-phase CML or disease resistance to interferon-α (INF-a) demonstrated a complete response [17,18,19]. As an inhibitor of the hallmark oncogenic driver of CML, the BCR-ABL fusion protein, a constituently active tyrosine kinase promoter of cell division expressed by the Philadelphia chromosome t(9:22), imatinib became the world’s first and prototypical targeted oncology agent [16,20].

Over the last twenty-plus years since the completion of the Human Genome Project, rapid advances in genomic analysis, particularly next-generation sequencing (NGS), have allowed for exponential growth in precision oncology, as the field of study since has been become known. Fundamentally, precision oncology can be defined as the molecular profiling of tumors to identify targetable genetic alterations by which deployment of the optimal therapy for a patient and their particular tumor can be achieved, principally through the discovery of somatic driver mutations arising from oncogenes and tumor-suppressor genes. The implementation of precision oncology is a highly coordinated effort between clinical oncologists, oncologic researchers, surgeons, radiologists, and pathologists. Basic challenges include radiographic diagnostic and procedural planning, acquisition of adequate tissue sample by means of surgery or image-guided techniques, and appropriate pathological examination. To date, genomic techniques such as next NGS and RNA sequencing have since identified countless single nucleotide polymorphisms, insertions and deletions (indels), rearrangements, gene fusions, and copy variants across many different malignancies, with hopes of identifying druggable targets [16,21]. As such, cancer treatment, which was once characterized by nonspecific cytotoxic therapies, is shifting toward the finer, pharmacogenomics-centered approach of precision oncology. Already, precision oncology has become standard of care in multiple cancers including nonsmall cell lung cancer (NSCLC), BRAF V600E mutated melanoma, fibroblast growth factor receptor 2 (FGFR2) rearranged cholangiocarcinoma, and others [22].

However, if it had once appeared that it would only be a matter of time before there would be molecular targets for all malignancies, and that, for a moment, the death of cancer seemed to be on the horizon, after two decades of tremendous breakthroughs and discoveries, precision oncology has failed to deliver on the promise of becoming a panacea to each and every malignancy. Numerous obstacles have been encountered in pursuit of the goal of precision therapy for many cancers, HCC in particular. Several of the complex challenges of further advancement include tumor evolution, intratumor heterogeneity, intolerance of multipathway inhibition, and lack of druggable targets [23]. Owing to the unexpected complexity that many tumors exhibit, some are of the opinion that precision oncology may never be able to deliver a silver bullet for all, perhaps even most cancers [19]. Despite this, continued efforts to improve and develop targeted therapies are underway.

Perhaps in no other malignancy are the shortcomings of current advances in precision oncology more apparent than in the treatment of HCC. Until recently, systemic therapy in HCC was limited to a single agent, sorafenib, which is not curative and initially demonstrated only a three-month median survival benefit. This meager gain is accompanied by scores of randomized clinical trials which failed to demonstrate benefit in HCC for a wide array of existing and experimental drugs [6,12]. Much of the difficulty in precision oncology within the realm of HCC treatment can be owed to the extreme genetic diversity of HCC and its molecularly heterogenous nature [24]. In truth, HCC can be thought of as an umbrella diagnosis that refers to what are, in effect, several different genetically and clinically distinct primary malignancies of the liver. Significant efforts to identify the primary mutagenic drivers of HCC and distinct subtypes of HCC have been ongoing since the inception of precision oncology. This research has created several new frameworks for clinical classification, but also shows that HCC is an increasingly complex disease and that much remains to be answered. The pathogenesis of HCC and the proposed main molecular pathways involved are described in the following section.

## 3. Pathogenesis of Hepatocellular Carcinoma

HCC is an incredibly complex and heterogenous malignancy. This fact has contributed greatly to the difficulty in characterizing the malignancy and in developing targeted precision oncology agents. HCC is believed to develop in a multistep oncogenic process, usually occurring in the context of a chronically inflamed, cirrhotic liver, in which low-grade dysplastic nodules (LGDN) progress to high-grade dysplastic nodules (HGDN) and then finally there is transformation to frank malignancy (Figure 1) [25,26,27]. This stepwise process is complex, and results from interplay among genetics, oncogenic risk factors including viral infections, toxin exposure, alcohol consumption, and crosstalk between tumor cells and an altered cellular microenvironment owing to concurrent cirrhosis and chronic liver inflammation [3,28,29]. The inflamed tumor microenvironment of the liver serves as substrate for the development of recurrent HCC primary tumors in a phenomenon termed the “field effect” [30].

The average HCC tumor is host to approximately 40 genomic aberrations. However, unlike many other cancers, it is believed that in HCC that few of these mutations serve as oncogenic drivers and that the majority of such mutations are merely “passenger” mutations [31]. Currently, it is thought that the accumulation of mutations is not entirely at random. There are likely several common pathways in which there are “trunk” mutations that form the core of HCC tumorigenesis, which are subsequently followed by “branch” mutations that are accumulated later in the disease course. Such a model was supported by a whole exome sequencing study of 34 tumors from 6 multifocal HCC patients, which mapped commonly mutated genes onto phylogenetic trees [32]. While oncogenesis is predominately through the accumulation of somatic mutations, conditions such as Li–Fraumeni syndrome, a germline TP53 mutation, can also predispose individuals to HCC development under rare circumstances [33,34].

Among aberrantly expressed genes and pathways that are suspected to be the main drivers in HCC, some of the most commonly found mutations include those in telomerase reverse transcriptase (TERT); WNT pathway activation; tumor protein p53 (TP53); Janus kinase/signal transducer and activator of transcription (JAK/STAT); RAS/mitogen-activated protein kinase (MAPK); phosphoinositide 3-kinases (PI3Ks)/AKT/mammalian target of rapamycin (mTOR); fibroblast growth factor 19 (FGF19); vascular endothelial growth factor (VEGF); nuclear factor kappa B (NF-kB) resulting in apoptosis evasion; and genes involving epigenetic modifications at the chromatin remodeling level, such as AT-rich interactive domain-containing protein 1A (ARID1A) [27,35,36,37].

A particular study performed whole exome sequencing of 243 tumors and identified 161 potential driver mutations, and ultimately 11 major pathways that were commonly mutated in at least 5% of tumors. Of the 11 pathways, the investigators found in descending order of frequency that the most commonly involved pathways were TERT promoter (60%), WNT/β-catenin (54%), PI3K/ATK/mTOR (51%), TP53/cell cycle (49%), MAPK (43%), hepatic differentiation pathway (34%), epigenetic regulation (32%), chromatin remodeling (28%), oxidative stress (12%), JAK/STAT (9%), and transforming growth-factor-β (TGF-β) (5%). On a mutational level, it was found that within the WNT/β-catenin pathway that the most mutated genes were *CTNNB1* (37.4%) and *AXIN1* (11.1%). Concerning the P53/cell cycle pathway, the most common aberrations were TP53 (24.3%), *CDKN21* (8.5%), and *ATM* (5.5%) [38]. Similarly, in chromatin remodeling and epigenetic modification, the most commonly mutated genes were *ARID1A* (12.8%), *ARID2* (6.8%), and *MLL2* (5.5%) [39]. Another large exome sequencing study, conducted by The Cancer Genome Atlas Research Network (TCGARN) in 2017, identified 26 “significantly mutated genes (SMGs)”, of which 18 genes had been previously identified as probable drivers of oncogenesis in HCC. These genes included *TP53* (31%), *AXIN1* (8%), *RB1* (4%), *CTNNB1* (27%), *ARID1A* (7%) chromatin remodeling, *ARID2* (5%) chromatin remodeling, *BAP1* (5%) chromatin remodeling, *NFE2L2* and *KEAP1*, as well as albumin (*ALB*) and *APOB* in 13% and 10%, respectively. Additionally, eight new potential drivers were identified, all of which being novel driver candidates in HCC and had also never been implicated in oncogenesis in any other cancer. These included *LZTR1* (3%), a gene that encodes a CUL3-containing E3 ligase complex, a gene already implicated in glioblastoma, and *EEF1A1*, a translation elongation factor [38].

Of the many proposed driver mutations, TERT promoter mutations have recently been under intense interest, owing to their high proportional representation in HCC (54–60%) and their unique role in oncogenesis [26,27,40,41,42]. Under normal circumstances, TERT is not expressed in human adult hepatocytes, which results in progressive telomere shortening with each subsequent cell division, cellular senescence, and ultimately cell death [42]. Reactivation of TERT and inappropriate expression of telomerase initiated malignancy in mouse models and is proposed to be an early mutational event in HCC development, as it is observed not only in frank HCC but also LGDN and HGDN. The mechanism of TERT reactivation is variable, but often includes promoter mutation (40–60%), followed by HBV insertion into the TERT promoter, activating the gene [43,44].

Other mechanisms of TERT activation include promoter amplification (5%) or promoter translocation (2–3%) [3,26,27]. Other proposed early mutational events include *VEGFA*, *ARID1B*, *KEAP1*, *TP53*, *MAP2K*, *NFKBIE*, *MET*, *STAT3*, and *MYC* [32], whereas activation of other frequently identified potential drivers, including TGF-β and WNT, are proposed to occur later [26].

## 4. Molecular Subclassification of HCC

With HCC demonstrating so much genomic variability, there have been extensive efforts to establish a uniform subclassification system with consistent molecular and prognostic features in order to better stratify disease treatment by grouping patients into distinct therapeutic classes. Thus far, attempts to classify HCC molecularly have yielded a variety of nonuniform classification systems that showcase the difficulty of the task and the great genetic diversity of HCC (Table 1). Broadly, HCC may be thought of has having two molecular classes, proliferation and nonproliferation [45]. The proliferation class accounts of 50% of all HCC, is HBV associated, with aberrations in TP53, RAS, mTOR, insulin-like growth factor (IGF), FGF19, and cyclin D1 (CCND1), and overall worse prognosis. On the other hand, the nonproliferation class, accounting for the other half of HCC, tends to be associated with alcohol and HCV and has improved outcomes relative to the proliferation class [3,27].

### 4.1. Lee Classification

An early attempt at classification by Lee et al. in a 2004 study was devised after analyzing complementary DNA (cDNA) microarrays of RNA extracted from 91 livers with HCC tumors in patient populations derived from China and Belgium. In this study, their transcriptome was compared to 19 morphologically normal livers [46]. This study demonstrated the expected genomic difference between HCC and normal livers together with the high genetic diversity of HCC, and it ultimately subdivided their cohort into two distinct subclasses based on survival: Cluster A (low survival) and Cluster B (higher survival). Cluster A was associated with higher α-fetoprotein (AFP) expression (62.5% vs. 42% in Cluster B), higher grade tumors (Edmonson grade III 77% vs. 50% in Cluster B), and a lower mean overall survival 30.3 +/− 8.02 months vs. 83.7 +/− 10.3 months for Cluster B [46]. Each cluster had a unique genetic signature, but Cluster A had a greater hazard ratio for carrying mutations associated with high rates of proliferation, including PNCA and cell cycle regulators CDK4, CCNB1, CCNA2, and CKS2, as compared with Cluster B [46]. The progenitor cell of HCC is debated, but thought that it can arise from adult hepatocytes and hepatocyte stem cells [47]. With two broad categories of HCC having emerged, each possessing distinct molecular characteristics, an attempt to determine if each category is derived from unique progenitor cells. Following their 2004 study, Lee et al. compared gene expression profiles of HCC with those of fetal mouse model hepatoblasts compared to adult hepatocytes, and found that while Cluster A had expression profiles comparable to both adult hepatocytes and fetal hepatoblasts, fetal hepatoblasts profile was exclusive to that cluster, suggesting that it may be the cell of origin [47].

### 4.2. Boyault Classification

Seeking to further elucidate the relationship between genomics and phenotype in HCC, a transcriptome microarray analysis using RT-PCR of 120 HCCs and 3 hepatocellular adenomas (HCA) derived from a French population yielded a classification system with six distinct subgroups (G1–G6) [48]. G1 was associated with low HBV copy number and overexpression of fetal liver genes. G2 demonstrated high HBV copy number and aberrations in the PIK3 and TP53 pathways. Both G1 and G2 were associated with AKT activation. G3 was associated with TP53 activation and cell cycle control dysregulation. G4, on the other hand, was a heterogenous group. G5 and G6 together were characterized by WNT and beta-catenin pathway activations [48]. Boyault et al. noted that their classification system was conducive to two overarching subgroups. Subgroup 1, comprising G1–G3, corresponds generally with the proliferation class, whereas Subgroup 2, consisting of G5 and G6 subclasses, is characterized by not only WNT activation, but were also not typically associated with HBV infection, which can be aptly characterized as being part of the nonproliferative class discussed by Llovet et al. [3]. Similarly, the proliferative and nonproliferative classes fit well with the classification system of Cluster A and Cluster B as originally proposed by Lee et al. [46].

### 4.3. Chiang Classification

A unique look at HCC tumors that were specifically HCV-associated produced a new and different classification schema which identified four subclasses: CTNNB1 activated, proliferation (IGF1R and ribosomal protein 6 (RPS6) phosphorylation enriched), Interferon-stimulated targets, and Polysomy 7 with upregulation of that chromosome’s respective genes [49]. This was performed by using a single nucleotide polymorphism array in addition to fluorescence in situ hybridization of tumor samples from 100 HCC transplant patients. The study was not powered to detect differences in survival between groups, but did identify that high-level enrichment of VEGFA amongst several study participants may serve as a drug target [49].

### 4.4. Hoshida Classification

Despite the fact that several independent attempts to create a diagnostic and prognostic classification system for HCC had resulted in similar systems, differences and a lack of uniformity remained. To address regional variability in HCC genetic profiles and reduce the bias introduced by the fact that no single standard genetic microarray has been used to identify molecular subclasses, yet another classification system was established as the result of a large meta-analysis by Hoshida et al., reviewing eight global patient cohorts. Their study observed three robust HCC subclasses, designated S1, S2, and S3 [50]. S1 was characterized by aberrant activation of the WNT pathway. Interestingly, rather than simply through mutation of the beta-catenin gene (*CTNNB1*), this pathway also saw TGF-B specific activation, which represented a new mechanism of WNT activation within HCC. S2 was characterized by MYC and AKT activation and saw higher AFP levels, corresponding with worse prognosis. Additionally, S1 and S2 showed greater nuclear accumulation of p53, resulting from inactivation mutations. S3 was deemed a hepatocyte differentiation class [50]. There was no histologic difference between S1 and S2 tumors, which suggests that they may represent two subclasses of the broader proliferation classification of HCC, while S3 corresponds more so with the nonproliferation class [3].

### 4.5. iCluster Classification

Even more recently, an integrated cluster analysis using whole exome sequencing of 363 HCC cases, in addition to DNA methylation, RNA and micro-RNA (miRNA), and proteome analyses, demonstrated three mutational signatures, which the authors designated as iClusters 1–3 [38]. iCluster 1 was associated with generally younger, Asian, female, normal weight, worse prognosis, and by Hoshida classification had the lowest fraction of differentiated cancers, corresponding to S2. Overall lower degree of CDKN2A mutation (32%) was observed when compared to iCluster 2 and iCluster 3 (69% and 63%, respectively). iCluster1 also saw lower TERT promoter mutation than the other clusters [38].

## 5. Early Attempts at Precision Therapy

Ultimately, the goal of each of the previously described classification systems is to provide a framework for HCC prognostication and help identify molecular targets for drug development [51]. For over a decade, sorafenib, a multi-kinase inhibitor with broad spread of targets, remained the only approved systemic therapy for HCC. A need for more targeted therapies emerged.

Over the past two decades, understanding of HCC genomics has improved, and the authors of each proposed classification system have identified potential oncologic targets deriving from the dominant pathways for each of the identified subclasses. The utility of these proposed targets could then be used to assign therapeutic and prognostic value to each subclass. For example, two pathways that are frequently implicated include WNT and ATK, in which activation mutations were identified in 50% of all tumors surveyed by Boyault et al. [48]. The WNT pathway has been demonstrated to be activated by both CTNNB1 mutations encoding beta-catenin (44%) and inactivation mutations of AXIN1 (<10%), and via accessory pathways involving TGF-β [50]. WNT may very well be an ideal molecular target, and dysregulation of the WNT pathway with CTNNB1 mutations are shown to be susceptible to modulation by sorafenib via reduced β-catenin levels and reduced levels of mRNA for known WNT target genes GLUL, LGR5, and TBX3 [52]. These findings suggested that WNT pathway inhibition could be potent, if physiologically tolerable. Another pathway that has been suggested as a potential pharmacologic target includes VEGFA, in which KDR/VEGFR-2 and FLT4/VEGFR-3 are among putative targets of sorafenib [49].

Although many putative targets of oncogenesis in HCC were identified, attempts to capitalize on these discoveries were frequently met with disappointing results. An ongoing challenge is that many of the most commonly mutated genes in HCC, such as TERT, TP53, and CTNNB1, are not druggable targets [53]. VEGF, which was noted to be a potential therapeutic target in the analysis by Chiang et al. (2008), was investigated in several trials assessing the role of VEGF antagonists in HCC. A phase II trial involving the VEGF inhibitor cedratinib was terminated prior to the second stage of the trial due to serious adverse events (SAEs) (5/17), principally grade III hypertension (29%) and cardiac ischemia, Every patient had at least one adverse event, which seemed to implicate every organ system. In particular were hyponatremia (65%) and hyperbilirubinemia (41%) [54]. In the same year, two separate phase III clinical trials using the combined VEGFR2/FGFR inhibitor brivanib were conducted. In a noninferiority study with overall survival as the primary endpoint, comparison between brivanib versus sorafenib as first-line treatment failed to reach its primary endpoint. Sorafenib demonstrated a higher overall survival of 9.9 months vs. 9.5 months for brivanib, and brivanib was less tolerable and had higher incidence of discontinuation due to adverse events (43% vs. 33%) for sorafenib [55]. Having not met primary endpoints as use for first-line therapy in patients who could not tolerate or subsequently failed sorafenib, a comparison between brivanib and best supportive care (BSC) versus BSC and placebo was conducted. This study, BRISK-PS, also, unfortunately, did not demonstrate a significant increase in overall survival in the brivanib arm compared to placebo (9.4 months vs. 8.2; hazard ratio [HR], 0.89; CI 95% 0.69–1.15; *p* = 0.3307) [56].

Trials seeking to exploit other proposed driver mutations in HCC, including EGFR, mTOR, and MEK, were similarly met with unsatisfactory results. Regarding EGFR, the SEARCH trial investigated the effect of sorafenib plus placebo versus sorafenib plus erlotinib with a primary outcome measure of overall survival. The trial demonstrated no significant difference in overall survival (9.5 vs. 8.5 months, respectively; hazard ratio [HR], 0.929; *p* = 0.408). Additionally, there was no significant difference in overall response rate or number of adverse events [57]. Similarly, the EVOLVE-1 trial in patients who either failed sorafenib or could not tolerate it was conducted using everolimus, an mTOR inhibitor. Everolimus, with BSC versus placebo in addition to BSC with overall outcome survival as primary endpoint, demonstrated no overall difference in survival with a median survival of 7.6 months in the everolimus arm versus 7.3 months with placebo (hazard ratio [HR], 1.05; CI 95% 0.86–1.27; *p* = 0.68). Similarly, there was no difference in overall time to progression; 3.0 and 2.6 months, respectively (hazard ratio [HR], 0.93; CI 95% 0.75–1.15) [58]. An attempt to probe the utility of inhibition of the RAS pathway, well known to be mutated in HCC, also failed to produce promising results. Specifically, two phase II clinical trials investigating the use of refametanib, a MEK inhibitor, showed that refametanib alone yielded an overall response rate (ORR) of 0% and overall survival of 5.8 months. In comparison to refametinib plus sorafenib, the ORR was 6.3% with an overall survival of 12.3 months. These studies were conducted in the context of a phase II trial conducted in Asia, BASIL, which used sorafenib/refametinib combination therapy and saw ORR of 75% in RAS mutated patients by circulating tumor DNA (ctDNA). However, both BASIL and the two phase II trials evaluated by Lim et al. demonstrated an overall prevalence of approximately 5% for RAS mutation in HCC. As such, the sorafenib/refametanib phase II conducted in the United States had comparable ORR for all study participants to that of BASIL, 6.3% and 6.9% [59,60]. These are just a small subset of the myriad efforts to apply the principles of precision oncology to HCC using the lessons learned from the genomic advances discovered at the turn of the millennium and the decade that followed, but in virtually all cases, the results failed to generate a significant breakthrough.

The reasons for why these approaches failed to produce significant results are varied. Certain targets, such as VEGF, EGFR, and mTOR, are associated with treatment-limiting adverse events, which either result in patient self-preferential discontinuation, SAEs necessitating a change in therapy, or simply a tenuous balance of risks and benefits [54,58]. Another reason for a lack of significant results may be many of these studies were simply not large enough to have sufficient power to detect differences, as most phase II trials enroll fewer than 50 patients. However, it may also be possible that the agents investigated lack sufficiently strong antitumor properties, as speculated with brivanib, which showed poor results as monotherapy and as combination therapy with sorafenib [56]. Even when agents demonstrated target effect, it also did not always mean clinical outcomes followed. Erlotinib in the SEARCH trial demonstrated tumor shrinkage and improved ORR, but this did not equate to improved overall survival [58]. Perhaps, however, the greatest challenge is the well-recognized genetic diversity and heterogeneity of HCC. Even when targeting a potential driver of oncogenesis in HCC, each potential driver mutation represents an overall small proportion of HCC patients [59,60]. Also, genes found to be frequently mutated in HCC and presumed to be truncal mutations with significant roles in carcinogenesis may in fact be less central and have a greater passenger role than previously thought [58].

### 5.1. Advances in Precision Therapy for Hepatocellular Carcinoma

#### 5.1.1. Multi-Kinase Inhibition

As of 2023, however, the treatment landscape for targeted therapies in HCC has rather significantly changed (Table 2). Ending sorafenib’s decade long reign as the sole systemic therapy approved for first-line treatment of advanced, nonresectable HCC, lenvatinib was approved in 2018 [14]. Similar to sorafenib, lenvatinib is a multi-kinase inhibitor that acts against VEGFR1-3, FGFR1-4, PDGFR-α, RET, and KIT [61]. Following promising signs of activity against HCC in a phase II clinical trial, lenvatinib was compared head-to-head with sorafenib in a phase III noninferiority trial as first-line therapy (REFLECT) with overall survival as the primary endpoint. Noninferiority was demonstrated with a median overall survival of 13.6 vs. 12.3 months with hazard ratio [HR]; 0.92, CI 95% 0.79–1.06 compared to sorafenib. Also demonstrated was improvement compared to sorafenib in all secondary endpoints, including progression-free survival, time to progression, and quality of life, as determined in part by the European Organization for Research and Treatment of Cancer Quality of Life Questionnaire C30 (EORTC-QLQ-C30) [61]. Additionally, a subgroup analysis looking at patients with objective response to lenvatinib versus nonresponders demonstrated that those with objective response had significantly longer median overall survival times (21.6 months vs. 11.9 months) when compared with nonresponders [61]. It was noted in the original study that those treated with lenvatinib with baseline AFP > 200 had lower overall survival than those with AFP < 200 [61]. It has been suggested that this imbalance in baseline prognostic factors was weighted against lenvatinib in the original study, and that after covariate analysis of the data from REFLECT suggest that when adjusted for differences in AFP at randomization, the efficacy of lenvatinib may actually be underestimated [62,63].

While sorafenib and lenvatinib remain the only two targeted small molecule first-line therapies approved by the USFDA for nonresectable HCC, advances have also been made in developing second-line therapies. The RESOURCE trial lead to approval of regorafenib as second-line therapy in those who progressed on sorafenib. Regorafenib, similar to sorafenib and lenvatinib, is a multi-kinase inhibitor of many angiogenic and pro-oncogenic kinases, including VEGFR1-3, tyrosine kinase with immunoglobulin-like and EGF-like domains 2 (TIE2), platelet-derived growth factor receptor (PDGFR-β), FGFR1, BRAF, RET, and KIT. With a primary endpoint of overall survival, RESOURCE demonstrated longer overall survival for regorafenib than placebo at 10.6 months vs. 7.8 months, respectively (hazard ratio [HR]; 0.63, *p* < 0.0001) [64]. Sequential multi-kinase inhibitors with partly overlapping target profiles may have additive effect, as it was noted that all primary and secondary outcomes in this trial for regorafenib were better than those of sorafenib when used as first-line therapy. Some of the beneficial effect of regorafenib is also speculated to be due to modulatory effect on surrounding inflammatory microenvironment of the tumor [64]. Another multi-kinase inhibitor approved for second-line use in HCC is cabozantinib, which has among its targets MET, VEGR1-3, KIT, and AXL, important drivers of oncogenesis and metastasis. Cabozantinib was approved given the results of the CELESTIAL trial, a phase III placebo-controlled trial in patients who had been previously treated with sorafenib and two or more prior systemic therapies. With overall survival as the primary endpoint, cabozantinib, when compared to placebo, had an overall survival of 11.3 vs. 7.2 months, respectively (hazard ratio [HR]; 0.70, CI 95% 0.55–0.88), in patients who only received sorafenib prior. Also shown to improve progression-free survival at 3.8 months for cabozantinib versus 1.8 months for placebo (HR 0.35, CI 95% 0.23 to 0.52) [65,66,67]. Finally, deviating from the other approved second-line targeted therapies is ramucirumab, a monoclonal antibody targeting VEGR2, which was approved after demonstrating effective antitumor activity in the phase III REACH trial without showing improvement in overall survival compared to placebo (9.2 vs. 7.6, hazard ratio [HR]; 0.87, *p* = 0.14), though it ultimately showed improved overall survival in the subgroup of patients with AFP > 400 (7.8 months vs. 4.2 months for ramucirumab versus placebo, respectively (hazard ratio [HR]; 0.674, *p* = 0.006 [68,69].

#### 5.1.2. Immune Checkpoint Inhibition

Progress in developing first- and second-line treatments has also made inroads with immunotherapies, first by targeting PD1 with nivolumab as well as with pembrolizumab. CheckMate 040, a phase I/II dose-escalation trial investigating nivolumab in advanced HCC, resulted in USFDA approval of the medication as second-line therapy as an outcome of promising ORR (20%) and durable response approved as second-line therapy [70]. Similarly, pembrolizumab received accelerated approval as second-line therapy as result of the KEYNOTE 224 trial, a phase II trial which demonstrated median overall survival of 12.9 months (CI 95% 9.7–15.5) for patients who had progressed on sorafenib [71]. Subsequently, KEYNOTE 240, a phase III, demonstrated a 13.9 vs. 10.6 median survival when compared to placebo (hazard ratio [HR]; 0.781, CI 95% 0.611–0.998; *p* = 0.0238), but did not meet its prespecified clinical endpoints. However, it confirmed the benefit of pembrolizumab by showing benefit with regard to overall survival, progression-free survival, and objective response rate [72].

Currently, there are many additional studies also seeking to assess the benefit of combination immune therapies as well [14]. One in particular, atezolizumab plus bevacizumab, actually demonstrated a higher overall survival and progression-free survival when compared to sorafenib when used as first-line therapy. Such was seen in the phase III IMbrave 150 trial, which showed a 12 month overall survival rate of 67.2% vs. 54.6% for atezolizumab/bevacizumab combination therapy and sorafenib, respectively (hazard ratio [HR]; 0.58; CI 95% 0.42–0.79; *p* < 0.0001). For the first time since the development of sorafenib has a clinical trial generated results which are better than the standard of care in first-line therapy for advanced HCC [73]. An additional combination immune checkpoint inhibitor therapy, nivolumab–ipilimumab, consisting of an anti-PD1 and an anti-CTLA antibody, respectively, saw approval for second-line use in HCC as result of subgroup analysis from the CheckMate 040 trial [74,75]. The most recent addition to the immune checkpoint inhibition armamentarium came with the results of the HIMALAYA trial. Presented in 2022, HIMALAYA demonstrated the superiority of combination therapy with tremelimumab–durvalumab over sorafenib as first-line therapy, with improved overall survival (hazard ratio [HR]; 0.78; 96% CI 96% 0.65–0.92; *p* = 0.0035) [76].

Repeated results showing clinical benefit with immune checkpoint inhibition demonstrates a strong class effect. Hence, immunotherapies are the new backbone of all ongoing research and appear to have greatest effect in combination, with approximately double the response rate and longer overall survival [77]. The improved response of combination therapy is likely due to synergistic effects of targeting multiple immune pathways and aiding in the prevention of drug resistance [78].

### 5.2. Matching Molecular Subclassification and Biomarkers with Systemic Therapy

What many successful small molecule therapies in both the first- and second-line settings have in common are that they are generally nonselective multi-tyrosine kinase inhibitors with a range of targets across different cellular pathways. This feature, by targeting multiple pathways, may help overcome primary and secondary resistance to more selective agents by addressing the core issue of tumor heterogeneity and extreme genetic diversity in HCC. As previously mentioned, immune checkpoint inhibitors show even greater therapeutic success than small molecule multi-kinase inhibitors. Large-scale genomics evaluations of HCC have generated several different molecular subclassifications, as previously explored in the sections above, and have also played a role in identifying commonly mutated pathways and potential therapeutic targets. Yet, while molecular subclassification may correlate with clinical features such as etiology and tumor behavior, it has not been successfully implemented as a tool in treatment selection [79]. Part of the issue lies in the fact that there are no reliable biomarkers in HCC on which to base clinical decision-making [80,81]. An exception is ramucirumab, which has been approved specifically for subsets of patients with AFP > 400 for use as second-line therapy [82]. In order to capitalize on the work and progress that has been made in better understanding the pathogenesis of HCC, studies need to be done to specifically evaluate for therapy effect when patients are stratified by subclass analysis. The current approach mainly focuses on attempting to exploit identified putative driver mutations by matching targeted therapies with those mutations, as identified in patients by NGS.

As such, clinical trials seeking to explore the use of biomarkers in therapy are underway. For example, there is a trial investigating fisogatinib, an FGFR4 inhibitor, in patients with FGF19 IHC + HCC, as FGF19 is the endogenous ligand for FGFR4, which is the most common of the fibroblast growth factor receptors in the liver [71,82,83]. New small molecule inhibitors of additional targets such as MET, PI3K-AKT-mTOR, and VEGF are also underway, as are trials seeking to identify associated biomarkers that may predict clinical response [71].

Aside from nonselective small molecule tyrosine kinase inhibitors, the other major class of emerging therapeutics is that of combination immune checkpoint inhibitors. This success is perhaps owing to their tumor agnostic nature, as they have already demonstrated utility across a variety of different cancers aside from HCC. Interestingly, however, PD-L1 status, which has been demonstrated to be a critical biomarker in the deployment of immune checkpoint inhibitors in other malignancies, has not been shown to be predictive of response in HCC [73]. An interesting consideration would be if the beneficial effect of immune checkpoint inhibitors can be attributed to the overall mutational burden of HCC and the high degree of molecular heterogeneity that it exhibits. Thus, the success of immunotherapy agents may overcome the fact that many identified driver mutations in HCC are not druggable targets. Additionally, they are less prone to drug resistance due to tumor heterogeneity and multiple cellular pathway aberrations. The role that tumor mutation burden plays in HCC is still uncertain and warrants greater investigation [57].

## 6. Review Limitations

This review is limited by the fact that while it presents a general oversight of the basic principles of HCC pathogenesis and its most common mechanisms, it is by no means exhaustive. As this review emphasizes, the genomic landscape of HCC is vast. There remain dozens of identified point mutations that were not discussed. Similarly, this review did not seek to explore the impact of the many indels, copy number variants, and chromosomal aberrations that have been identified, nor did this review seek to provide extensive discussion of mechanisms of HCC recurrence, i.e., “early” vs. “late” recurrence. This review also did not elaborate on the role of germline mutation in HCC pathogenesis. Another limitation, regarding discussion of the development of targeted agents for HCC, is that this paper is also not fully comprehensive. There remain scores of clinical trials that have been performed and were not mentioned. Instead, presented here is a small sample of specific attempts to develop therapies for what appeared to be promising molecular targets.

## 7. Conclusions and Future Directions

In summary, HCC remains one of the most common malignancies in the world and is responsible for a significant portion of cancer-related mortality. For the better half of the first decade of the new millennium, HCC lacked systemic therapeutic options owing to the difficulty in developing precision oncologic targets for what research has subsequently discovered is an extremely diverse and heterogenous disease. Even with improved understanding of the genomics and pathogenesis of HCC, many purported drivers of HCC were found to be unactionable [84].

A recent explosion of new first- and second-line therapies, however, largely involving immune checkpoint inhibition, have emerged over the course of the last five years, and with many promising clinical trials underway, new approvals are likely to come. This has been accompanied by significant improvement in discovering the main molecular and pathogenic mechanisms of HCC. As efforts to define and classify subclasses of HCC have made progress, the task remains incomplete, as no universal classification system for HCC has been recognized, nor has there been success in matching the identified subclasses of HCC to specific classes of therapy.

The challenge remains to fully integrate the progress made in HCC genomics and molecular subclassification and the gains being made in targeted pharmaceutical therapies to establish a diagnostic schema for HCC that has both prognostic value and therapeutic implications. Ultimately, the goal is for each subclass to have matched antineoplastic agents. Therefore, further large-scale genomics testing of diverse, global populations of individuals with HCC, derived from multiple etiologies, should be undertaken to identify useful patterns of gene mutation. Specifically, greater attention should be given to identifying favorable prognostic features regarding the use of immune checkpoint inhibitors beyond PDL1 status and tumor mutational burden.

## Figures and Tables

**Figure 1 ijms-24-13274-f001:**
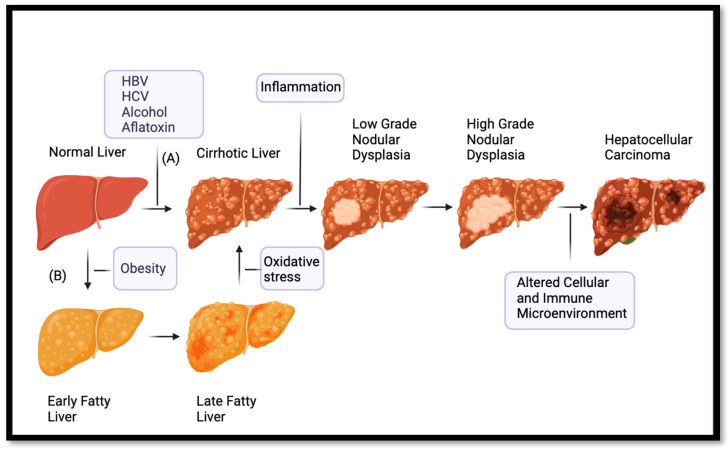
Simplified diagram outlining the main mechanisms of pathogenesis of hepatocellular carcinoma (HCC). In pathway (A), a morphologically normal liver is exposed chronically to pro-cirrhotic viruses or toxins. The most common risk factors are hepatitis B virus (HBV), hepatitis C virus, (HCV), alcohol, and aflatoxin. This exposure results in cirrhosis, which over time through chronic inflammation promotes the formation of low-grade dysplastic nodules (LGDN), which then progress to high-grade dysplastic nodules (HGDN). The altered cellular and immune environment of the cirrhotic liver causes accumulation of driver and passenger mutations and finally the development of frank HCC. In pathway (B), an alternative and growing share of HCC development occurs through fatty liver disease, which generates more oxidative stress as the disease progresses. This oxidative stress then leads to cirrhosis by which, similar to pathway (A), chronic inflammation promotes the development of LGDN that transform into HGDN and eventually HCC due to similar alterations of the immune and cellular environment of the cirrhotic and chronically inflamed liver. Created using Biorender.com.

**Table 1 ijms-24-13274-t001:** Major classification systems of hepatocellular carcinoma (HCC). The table is organized with rows corresponding to each of the major classification systems. Each system’s subclassifications (clinical features and mutations) are associated with each column. Rows are in descending order, based on the classification system’s year of origin, with the oldest at the top. Columns represent the two overarching categories of HCC and consist of corresponding subclassifications across classification systems and shared clinical features and mutations.

Classification System	Subclass(es)	
Major Subclasses	Proliferative	Nonproliferative
Lee 2004	Cluster A	Cluster B
Boyault 2006	G1, G2, G3	G4, G5, G6
Chiang 2008	Proliferation	CTNNB1, Polysomy 7, Interferon
Hoshida 2009	S1, S2	S3
TCGARN 2017	iCluster1, iCluster2	iCluster3
Clinical Features	Poor survival, high vascular invasion, women, Asian, high AFP, HBV, normal body weight, poorly differentiated hepatocyte	Improved survival, low vascular invasion, alcohol, HCV, lower AFP, well-differentiated hepatocyte, smaller tumor size
Mutations	CDK4, CNNB1, CKS2, PTMA/ProT, SET, MAPK3, CCNB1, CCNA2, AXIN1, PIK3CA, TP53, AKT, MYC, TGF-β, TERT, MYBL2 PLK1, MK167	CTNNB1, CDH1, TCF1 CDKN2A silencing by hypermethylation; Relatively high TERT promoter mutation TP53

**Table 2 ijms-24-13274-t002:** Current USFDA-approved first- and second-line systemic therapies for nonresectable HCC. Drugs shown include corresponding drug targets, clinical trial prompting approval with national clinical trial (NCT) number, primary endpoints investigated, and year of approval.

Therapy	Line of Treatment	Target (s)	Trial Name (National Clinical Trial Number)	Primary Endpoint (s)	Year
Sorafenib	First	VEGFA, KDR/VEGFR-2, FLT4/VEGFR-3	SHARP (NCT00105443)	Overall Survival Time to Symptomatic Progression	2007
Regorafenib	Second	VEGFR1-3, TIE2, PDGFR-β, GFGR1, BRAF, RET, KIT	RESOURCE (NCT01774344)	Overall Survival	2017
Nivolumab	Second	PD1	CheckMate 040 (NCT01658878)	Safety Tolerability Overall Response Rate	2017
Lenvatinib	First	VEGFR1-3, FGFR1-4, PDFR-α, RET, KIT	REFLECT (NCT01761266)	Overall Survival	2018
Pembrolizumab	Second	PD1	KEYNOTE 224 (NCT02702414) KEYNOTE 240 (NCT02702401)	Objective Response Rate Overall Survival Progression Free Survival	2018
Cabozantinib	Second	AXL, FLT-3, KIT, MER, MET, RET, ROS1, TIE-2, TRKB, TYRO3, VEGFR1-4	CELESTIAL (NCT01908426)	Overall Survival	2019
Ramucirumab	Second	VEGR2	REACH (NCT01140347)	Overall Survival	2019
Atezolizumab–Bevacizumab	First	PD1-VEGF	IMbrave 150 (NCT03434379)	Overall Survival Progression Free Survival	2020
Nivolumab–Ipilimumab	Second	PD1-CTLA	CheckMate 040 (NCT01658878)	Safety Tolerability Overall Response Rate	2020
Darvalumab–Tremilimuab	First	PDL1-CTLA4	HIMALAYA (NCT03298451)	Overall Survival	2022

## Data Availability

Not applicable.

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
