# Peer review of "Hepatocellular Carcinoma: Past and Present Challenges and Progress in Molecular Classification and Precision Oncology"

_ijms, 2023, doi:10.3390/ijms241713274_

Round 1

Reviewer 1 Report

General Comments:

The manuscript entitled " Hepatocellular carcinoma: Past and present challenges and progress in molecular classification and precision oncology

is trying to provide the evidence of Past and present challenges and progress in molecular classification and precision oncology. After reviewing this manuscript, I do have some concerns about the research.

May concern is about the citation is not common using roman numerals; please confirm to template to the journal.

Author also should elaborate the role of germline variants driving cancer development beside the somatic mutation

What is the solution for undruggable drug target genes as author mentioned has many challenges?

Point 4. Pathogenesis of Hepatocellular Carcinoma; author better to using figure to elaborate the pathogenesis of HCC

Table 1 seems to be picture, it does not present good visualization data, besides, it is difficult to understand. Author should modify it and reorganized using good visualization

Author should mention the abbreviation of content of Table 2. Such as NCT numbers and soon

.

Point 8. Integrating Genomics and Precision Oncology; reviewer recommend to use good vsiualization eg using CANVA or Biorender etc

Author should explain the innovations of this study?

Author should explain the "limitations" of the study.

Author Response

  1. May concern is about the citation is not common using roman numerals; please confirm to template to the journal.

Reply: This has been addressed. All numbers in Arabic numerals.

  1. Author also should elaborate the role of germline variants driving cancer development beside the somatic mutation

Reply: The primary mechanisms of pathogenesis of HCC involved chronic exposure of oncogenic viruses (HBV and HCV), chronic alcohol consumption, NASH/NAFLD, as well as less commonly aflatoxin exposure. Germline predisposition has been recorded in HCC pathogenesis but remains a minor mechanism which is beyond the scope of the general overview of HCC pathogenesis. A brief mention regarding germline mutation as a minor mechanism and example of germline TP53 mutations has been included.

  1. What is the solution for undruggable drug target genes as author mentioned has many challenges?

Reply: I have elaborated that immune checkpoint inhibition may provide a work around to this problem as they have demonstrated good effect against HCC despite tumors possessing. multiple cellular pathway aberrations. The other solution, as detailed in a section of the manuscript, would be development of new pharmaceuticals targeting these yet undruggable targets, though extensive efforts have failed, and now clinical trials are focused more on combination immunotherapies.

  1. Pathogenesis of Hepatocellular Carcinoma; author better to using figure to elaborate the pathogenesis of HCC

Reply: Figure Created using Biorender to demonstrate the general pathogenesis of HCC

  1. Table 1 seems to be picture, it does not present good visualization data, besides, it is difficult to understand. Author should modify it and reorganized using good visualization

Reply: Agreed that the first  iteration of the table was difficult to understand and  had poor visualization, table has been re-done to be in significantly simplified format.

  1. Author should mention the abbreviation of content of Table 2. Such as NCT numbers and so on

Reply:  Abbreviation of NCT explained. Table updated with new information.

  1. Integrating Genomics and Precision Oncology; reviewer recommend to use good visualization eg using CANVA or Biorender etc

Reply: Please see above response regarding using Biorender to create new figure for pathogenesis of HCC

  1. Author should explain the innovations of this study?

Reply: The purpose of this review is to provide an integrated narrative account of the simultaneous efforts to establish a uniform and validated molecular sub-classification system for HCC with prognostic and therapeutic utility with the ongoing but separate pursuit of identifying systemic therapeutic agents.

  1. Author should explain the "limitations" of the study.

Reply: In the conclusion I have listed the following limitations

  1. Not a comprehensive discussion of the molecular landscape and pathogenesis
  2. No detailed discussion of HCC recurrence
  3. No detailed discussion of the role of germ-line mutations beyond their mention
  4. Not a comprehensive recounting of the scores of clinical trials conducted.

Reviewer 2 Report

In this review, the authors provided a detailed narrative account of the unique challenges faced in hepatocellular carcinoma (HCC) with regard to efforts to categorize HCC prognostically and therapeutically according to different molecular subclassifications. In addition,  they summarized past and present attempts at implementing targeted systemic therapies, and the use of clinical biomarkers in identifying potential therapies of benefit and the elusive ideal of uniting each patient’s unique HCC with a targeted systemic therapy.

Specific comments

Introduction Many review articles on similar topics have been published, and the authors need to explain the necessity and innovation of this paper. e.g. J Hepatol. 2020 Feb;72(2):215-229; Acta Gastroenterol Belg. 2020 Apr-Jun;83(2):309-312.

The structure of the article is somewhat loose, and the logic needs to be strengthened. It is suggested that the second part and the third part of the text can be integrated.

A diagram is suggested to summarize the key pathological mechanisms involved Pathogenesis of Hepatocellular Carcinoma.

P7, The table is irregular and should be displayed with a three-line table.

In the fifth part, please set subheadings to make it more organized.

Conclusions More certainty is needed to summarize current research progress of the molecular classification and precision oncology of hepatocellular carcinoma, and the clinical application needs to be prospected.

  • The logic between different modules of the text needs to be strengthened.

Author Response

  1. Introduction Many review articles on similar topics have been published, and the authors need to explain the necessity and innovation of this paper. e.g. J Hepatol. 2020 Feb;72(2):215-229; Acta Gastroenterol Belg. 2020 Apr-Jun;83(2):309-312.

Reply: The purpose of this review is to provide an integrated narrative account of the simultaneous efforts to establish a uniform and validated molecular subclassification system for HCC with prognostic and therapeutic utility with the ongoing but separate pursuit of identifying systemic therapeutic agents.

  1. The structure of the article is somewhat loose, and the logic needs to be strengthened. It is suggested that the second part and the third part of the text can be integrated.

Reply: Second and third sections integrated into a single section

  1. A diagram is suggested to summarize the key pathological mechanisms involved Pathogenesis of Hepatocellular Carcinoma.

Reply: Figure was created using BioRender

  1. The table is irregular and should be displayed with a three-line table.

Reply: This has been corrected

  1. In the fifth part, please set subheadings to make it more organized.

Reply: Subheadings set

  1. Conclusions More certainty is needed to summarize current research progress of the molecular classification and precision oncology of hepatocellular carcinoma, and the clinical application needs to be prospected.

Reply: Further direction in research regarding the need to investigate favorable prognostic features in the use of immune checkpoint inhibition has bene proposed as well as the need for addition high powered large genomics studies

Comments on the Quality of English Language

  1. The logic between different modules of the text needs to be strengthened.

Reply: Reorganization with sub headers and additional explanation of the purpose of each section have been added to facilitate flow of manuscript

Round 2

Reviewer 1 Report

The authors have answered each question though further justified clarification in the manuscript. From my side, it can be accepted.

Reviewer 2 Report

The manuscript is improved. Please provide a brief explanation about molecular subclassification of HCC.

It is advised to polish the English, please follow the grammar rules while constructing long sentences.
